# Chemical and Phytocoenological Characteristics of Two Different Slovak Peatlands

**DOI:** 10.3390/plants10071290

**Published:** 2021-06-24

**Authors:** Danica Fazekašová, Gabriela Barančíková, Juraj Fazekaš, Lenka Štofejová, Ján Halas, Tadeáš Litavec, Tibor Liptaj

**Affiliations:** 1Department of Environmental Management, Faculty of Management, University of Prešov, Konštantínova 16, 080 01 Prešov, Slovakia; danica.fazekasova@unipo.sk (D.F.); gabriela.barancikova@unipo.sk (G.B.); lenka.stofejova@smail.unipo.sk (L.Š.); 2National Agricultural and Food Centre—Soil Science and Conservation Research Institute, Bratislava, Regional Working Place, 080 01 Prešov, Slovakia; jan.halas@nppc.sk (J.H.); tadeas.litavec@nppc.sk (T.L.); 3Faculty of Chemical and Food Technology, Slovak University of Technology in Bratislava, 812 37 Bratislava, Slovakia; tibor.liptaj@stuba.sk

**Keywords:** bog, fen, plants, organic carbon, pH

## Abstract

This paper presents the results of pedological and phytocoenological research focused on the detailed research of chemical parameters (pH, organic carbon, and nutrients), risk elements (As-metalloid, Cd, Co, Cr, Cu, Ni, Pb, and Zn), and species composition of the vegetation of two different peatlands on the territory of Slovakia—Belianske Lúky (a fen) and Rudné (a bog). Sampling points were selected to characterize the profile of the organosol within the peatland, the soil profile between the peatland and the agricultural land, and the soil profile of the outlying agricultural land, which is used as permanent grassland. Based on phytocoenological records, a semi-quantitative analysis of taxa in accordance with the Braun–Blanquet scale was performed. The study revealed that the thickness of the peat horizon of the fen in comparison with the bog is very low. In terms of the quality of organic matter, the monitored peatlands are dominated by fresh plant residues such as cellulose and lignin. Differences between individual types of peatlands were also found in the soil reaction and the supply of nitrogen to the organic matter of peat. The values of the soil exchange reaction were neutral on the fen, as well as slightly alkaline but extremely low on the bog. A significantly higher nitrogen supply was found in the organic matter of the fen in contrast to the bog. At the same time, extremely low content of accessible P and an above-limit content of As in the surface horizons were also found on the fen. From the phytocoenological point of view, 22 plant species were identified on the fen, while only five species were identified on the bog, which also affected the higher diversity (*H’*) and equitability (*e*). The results of the statistical testing confirmed the diversity of the studied peatlands and the different impact of environmental variables on plant diversity.

## 1. Introduction

Peatlands are among the rarest ecosystems in the world and have great importance for the protection of global biodiversity [1,2], the minimization of flood risks [3], and the mitigation of climate change [4]. Peatlands are ecosystems that are created on locations permanently waterlogged by precipitation, surface, or underground water. Understanding the hydrological system is crucial for sustainable land development, and effective soil and nature conservation and site hydrology appear as the main force of secondary succession [5,6,7]. In conditions of reduced air, organic residues in various stages of decomposition are accumulated, and peatland is created. Peat consists of more than 50% of combustible organic matter in dry matter [8,9]. A very brief characterization of peatland was proposed by Lindsay [10]: peatland is wetland covered by peat vegetation. In many well-developed peatlands, peat depths can exceed two metres [11]. Peatlands are divided into two main types—bogs and fens—based on their hydrological, bioclimatic, and chemical conditions, as well as nutrient and flora status [12,13].

In central Europe, several types of peat can be found: oligotrophic bogs, oligo-eutrophic fens, polytrophic fens, and eutrotrophic-polytrophic base-rich fens [14]. Three types of peatlands can be distinguished in Slovakia: raised bogs, transitional mires, and poor or rich fens. Slovak peatlands represent quite a small area, estimated at 26,000 ha, which is 0.57% of the total area of Slovakia [15]. Fens are rare and vulnerable ecosystems [16,17,18,19]. Diggelen et al. [20] presented a framework for the conservation and restoration of these ecosystems. This consists of optimizing abiotic conditions, safeguarding the propagule availability of the target species, creating and maintaining conditions for the (re)establishment of these species, and appropriate management to keep the conditions suitable. A fen peat accumulation may be dominated by sedge, reed, shrub, or forest [21].

The largest fen in Slovakia is the nature reservation Belianske Lúky, located in the High Tatras region. The Belianske Lúky fen arose more than 10,000 years ago, making it one of the oldest, largest (approx. 100 ha), and best-protected fens of European importance in Slovakia [22]. Peat has gradually accumulated over the ages and its maximum depth reaches up to 2.8 m. The Belianske Lúky fen is characterized as a calcareous fen of the transitional type, typical of the southern foothills of the High and Belianske Tatras [23,24]. Since 2003, the Belianske Lúky fen has been a national nature reservation. Belianske Lúky is classified by the NATURA 2000 system among the Territories of European Importance [25].

A fragment of the unique type of highland peat bog in Slovakia is the nature reserve Rudné near Suchá Hora village in the territory of Upper Orava. Rudné represents a very valuable fragment of actively raised bog habitat. Rudné is a very valuable fragment of the active habitat of the elevated swamp. Climatic conditions in connection with the impermeable subsoil of flysch rocks were the basic building blocks in the formation of large peatlands in the northern part of Orava. The specific geological and climatic conditions of this area in the north of Slovakia conditioned the formation of peat bogs, the so-called Orava forests. They have the character of bogs, degraded bogs, and transitional peat bogs. However, some larger complexes were drained and rich deposits of peat were extracted [22,26,27]. The Rudné bog is unique not only in Slovakia but also in the whole of Central Europe, confirming the occurrence of rare and endangered species of fauna and flora [22,28].

The aim of this study is to identify indicator plant species of two different peatlands in Slovakia—the Belianske Lúky fen and the Rudné bog as well as the surrounding areas—and to examine the effects of environmental variables (organic carbon content (OC), total nitrogen (Nt), soil reaction (pH / KCl), available phosphorus (P), available potassium (K), and selected risk elements (As, Cd, Co, Cr, Cu, Ni, Pb, and Zn) for vegetation composition, diversity, and equitability.

## 2. Materials and Methods

### 2.1. Characteristics of the Studied Sites

The Belianske Lúky fen (altitude 49°13.058′, longitude 20°23.002′) is located in the Poprad part of Spišská Kotlina Basin in the cadastre of the town of Spišská Belá at an altitude of 670–695 mamsl (Figure 1). According to the geomorphological division, the area belongs to the Podtatranská Kotlina Basin [29], built by the Inner Carpathian flysch. It includes basal conglomerates and shale claystones, while sandstones are added towards the overburden. The alternation of these layers creates a typical intra-Carpathian Paleogene that originated in the Tertiary. The Belianske Lúky bog is located in a cold, very humid climatic area, with an average temperature of 14–16 °C in July and −5 °C in January and an annual rainfall of 600–750 mm [30]. The hydrological area is drained by the river Poprad with tributaries [31].

Rudné Nature Reserve (bog) (altitude 49°23.671′, longitude 19°46.839′) is located in the north-eastern part of Orava in the Oravská Kotlina Basin, at an altitude of 740 mamsl. From the geological point of view, the Oravská Kotlina Basin belongs to the flysch zone and was built by late Tertiary sediments, such as clay, sand, and gravel. Basin uplands are the basic type of topography. The locality of the Rudné bog in the cadastral area of the village of Suchá Hora belongs to a cold, very humid climatic area. The average temperature in July reaches values between 12 and 14 °C, and −5 to −6 °C in January. The annual total precipitation is at the level of 800–900 mm [30].

### 2.2. Soil Sampling and Analyses

After field reconnaissance, sampling sites were selected to characterize the profile of the organosol within the peatland, the soil profile between the peatland and the agricultural land, and the soil profile of the outlying agricultural land that is used as permanent grassland. Plant vegetation cover helped in the selection of the location of the probe. The Belianske Lúky fen was classified as gleyic and fenny Organosol. The soil at the edge of the fen is classified as organogenic Rendzina. Agricultural land outside the fen in the Belianske Lúky locality is represented by modal, granular-clay Rendzina (Figure 2). The sampling depth was chosen based on visual (morphological) changes in the soil profile. At the Belianske Lúky site, peat samples were taken from the following depths: 5–15, 20–30, and 110–120 cm. The fen profile was filled with lateral water immediately after digging. Samples from the subsoil were taken with the help of a soil drill. The edge of the fen at the Belianske Lúky site was 95 m away from the fen point and the agricultural land was 118 m away from the fen sampling point.

The Rudné bog was classified as gleyic Organosol, the soil on the edge of the bog as pseudogleyic Cambisol, and the agricultural soil outside the bog is represented by the soil representative modal, granular-clay Pseudogley [32] (Figure 3). At the Rudné locality, samples were taken from a depth of 0–10 cm, 15–25 cm, 35–45 cm, and 95–103 cm. From the edge of the peat, as well as from the agricultural land near the Rudné bog, samples were taken only from the upper horizon, as it was a very shallow soil. A sample was taken from the edge of the bog from the depth of 5–10 cm and agricultural soil from the depth of 3–13 cm. The edge of the bog in the Rudné locality was 145 m away from the point on the bog and the agricultural land is 177 m away from the sampling point of the bog.

At individual depths, we monitored the content of organic carbon (OC), total nitrogen (Nt), soil reaction (pH / KCl), available phosphorus (P), available potassium (K), and selected risk elements (As-metalloid, Cd, Co, Cr, Cu, Ni, Pb, and Zn). Total carbon and nitrogen were determined on a Euro EA 3000 elemental analyzer in CN configuration (Eurovector Instruments & Software, Milano, Italy). The OC content was calculated from the total carbon content after correction for carbonates. The soil reaction was determined as the exchange soil reaction in 1 M KCl neutral salt solution (20 g of soil mixed with 50 mL of 1 M KCl) potentiometrically on a Mettler Toledo pH-meter (Mettler-Toledo Group, 8603 Schwerzenbach, Switzerland). The available phosphorus was determined by extracting the soil with Mehlich III extraction solution by atomic absorption spectrometry (AAS) by VARIAN AA240FS, AA240Z + GTA120 spectrometer (Varian, Melbourne, Australia). Accessible potassium and risk elements (As, Cd, Co, Cu, Cr, Ni, Pb, and Zn) were determined by segmented flow analysis (SFA) on a SKALAR SAN Plus instrument (Skalar analytical B.V., Breda, The Netherlands). To determine the detailed chemical structure of peat organic matter, detailed characterization of the organic matter in the upper horizon of peatlands was determined by the method of nuclear magnetic resonance carbon ^13^C in the solid state (high-resolution solid-state ^13^C NMR spectra) by spectrometer Varian/Agilent VNMRS 600 MHz (Agilent Technologies, Santa Clara, CA, USA). The methods for determination of the selected soil characteristics were selected in accordance with the methodology of Kobza et al. [33].

Semiquantitative analysis of taxa is presented in the samples taken from the studied area. Based on phytocoenological records, the botanical composition of peatlands was determined. The size of the area was 16 m^2^. The species composition of vascular plants by floor and cover was recorded in the record. Present species were evaluated in accordance with the Braun–Blanquet scale (5—cover of 75 to 100%, 4—cover of 50 to 75%, 3—cover of 25 to 50%, 2—cover of 5 to 25%, 1—under cover of 5%, +—cover of negligible, r—occasionally). Nomenclature is in accordance with Marhold and Hindák [34]. The species diversity of the communities of the studied localities was evaluated by the Shannon diversity index (*H’*):$$H’=-\overset{s}{\underset{i=1}{\sum }}\frac{xi}{N}log2\frac{xi}{N}$$
where *s*—the number of species, *xi*—individuals of one particular species found, *N*—divided by the total number of individuals found, *i*—1 [35]. The results were evaluated based on scales: 1 extremely low (< 0.5), 2 very low (0.5–1), 3 medium-low (1–1.7), 4 low (1.7–2.5), 5 low to moderate (2.5–3.3), 6 medium (3.3–4), 7 moderately high (4–5), 8 high (5–7), 9 very high (7–10), and 10 extremely high (>10) [35]. The composure of the communities was evaluated by the Equitability Index (*e*):$$e=\frac{H\prime }{log2s}$$

The closer the values are to 1, the more balanced is the community [36]. Hierarchical cluster analysis (HCA) and non-metric multidimensional scaling (NMDS) were performed using PAST 4.

## 3. Results and Discussion

### 3.1. Organic Carbon Contents

Data of the soil organic matter (SOM) content can be used for the assessment of stocks and changes of organic carbon (OC) in peatland soils, which is also pointed out in the study of Klingenfuß et al. [37]. In the peatlands we studied, a considerably higher amount of organic carbon (OC) in the upper horizon was found on the bog Rudné and a lower amount of the organic matter contains the fen Belianske Lúky (Figure 4). Moreover, significant differences in organic carbon contents were found in the profile of both the studied peats. On the bog Rudné, the depth of peat horizon was up to 50 cm; in 1 m depth, only the minimal contents of OC were determined. A different situation was found on the fen Belianske Lúky. The real peat was found only in the first 10 cm. In deeper horizons, <50% of the organic matter was found (Figure 4). The gradual decrease in the organic matter was observed not only in the depth of peat profile but also within the distance from the peat (Figure 4 and Figure 5). Different organic carbon contents between the Rudné and the Belianske Lúky pastures can be explained by different soil types. The Belianske Lúky pasture soil type is Rendzic Leptosoil, which has a characteristically higher OC content in comparison with Haplic Stagnosol—the soil type of the Rudné pasture.

The detailed chemical structure of organic carbon in the upper peat horizon was measured for bogs and fens by the high-resolution ^13^C nuclear magnetic resonance technique (high-resolution ^13^C NMR spectra) [38,39]. The most distinct peak in the ^13^C NMR spectrum of the Rudné bog (Figure 6) and the Belianske Lúky fen (Figure 7) is a significant peak in the 75 ppm region, which is characteristic of polysaccharides [40] and indicates that part of the cellulose and other carbohydrates which were not broken down during the peat formation process [41]. The 160–100 ppm region (Figure 7) contains aromatic carbons that may be part of the lignin, or humic substances, which are an important part of the organic matter of peat [41,42]. A clear peak in the range of 120–145 ppm is characteristic of non-protonated aromatic carbons [41]. A relatively broad peak in the 175 ppm region is characteristic of carbon carboxyls, amides, and carbonyl groups [40]. The aromatic area of the Rudné bog is relatively indistinct (Figure 6).

Based on the measured data (Table 1), we can state that aliphatic carbons dominate in the organic matter of peat on the Rudné bog, similarly to fens, their percentage is higher on the bog than on the fen. The higher proportion of aliphatic structures over aromatic ones is also reflected in the degree of aromatization, which is lower on the bog than on the fen. Compared to the organic matter of Slovak soils, where the values of the degree of aromatization (α) are in the range of 34–60% [43,44], the values of the evaluated peat are mainly low on bogs.

### 3.2. Contents of Nutrients, Soil Reaction, and Risk Elements Contents

The concentration of the total nitrogen (Nt) on the Rudné bog is relatively low compared to the Belianske Lúky fen, ranging from 12,000–14,500 mg/kg (Table 2). These findings are consistent with Davis and Anderson [45] and Szajdak et al. [46], who report high concentrations of Nt on fens and extremely low on bogs. On the Rudné bog, the concentration of Nt is fairly balanced similarly to the OC concentration in peat profile up to 50 cm. The high concentration of OC and low Nt on the Rudné bog is reflected in very high values of the C/N parameter. An extremely high C/N value, mainly in the first horizon of the Rudné bog, indicates extremely low nitrogen stock on the bog organic matter. High values of Nt in comparison with OC on the Belianske Lúky fen are reflected also in low values of the C/N parameter and indicate significantly higher nitrogen stock on the fen in comparison with the bog. As opposed to the total nitrogen, which was higher on the fen (Belianske Lúky locality), the values of the available phosphorus and potassium are significantly higher on highland bog type peat (Rudné locality). In particular, an extremely low figure is an available P on the fen but also soil on the Belianske Lúky locality. Potassium content on the bog is higher in comparison with the fen; however, a higher content of K in the soil was found on the Belianske Lúky locality than on the Rudné locality (Table 2). The data for N, K, and C/N are consistent with the research of Bhuiyan et al. [47], which found values for bog and fen at a level comparable with our values.

Significant differences were found in pH values, closely related to the peat type. On the highland bog Rudné, extremely low pH values were observed throughout the whole soil profile. On the fen Belianske Lúky, the pH values were neutral or slightly alkaline. The measured pH values are in agreement with the literature data showing pH ranges within 4–8 on fens and within 3–4.5 on bogs [45,48]. Values of pH on transition and agricultural soils are similar to their competent peats (Table 2).

Risk elements contents on both the studied peats (the bog and the fen) and the pasture near the peat locality are quite low. Only on the Belianske Lúky fen, a very high concentration of arsenic was found in the first two horizons, whereas on the Rudné bog, a significantly higher concentration of cadmium was determined in the depth 15–25 cm (Table 3). The above-limit concentration of Cd in the horizon of 15–25 cm can come from anthropogenic pollution—mainly from energy, industrial activity, and transport. Norton et al. [49] report in their study that in the lake and ombrotrophic peatland sediments, the rate of Cd accumulation increased two to three times above background levels, beginning about 100 years ago. In accordance with the above findings, Martinez-Cortizas et al. [50], confirmed in their work increased concentrations of Zn and Cd in the upper parts of the peat bog with small changes as regards the depth, but Pb showed significant peaks in the deeper layers.

Belianske Tatras carbonate minerals can contain a high concentration of As. After weathering and migration to groundwater, arsenic can be immobilized in the surface layer of peat because the peat organic matter tends to bind risk elements [51]. The findings of Hoffman et al. [52] provide spectroscopic evidence for two yet unconfirmed As(III)–NOM (natural organic matter) interaction mechanisms, which may play a vital role in the cycling of As in sub- and anoxic NOM-rich environments, such as peatlands, peaty sediments, swamps, or rice paddies.

### 3.3. Phytocoenological Characteristics

Plants are the most important functional species group in peatland ecosystems. Peatland plants form their growth substrate i.e., peat. Plant communities also greatly affect biodiversity at the ecosystem and the landscape level [53]. Phytocoenological data of peatlands of plant communities in the Belianske Lúky and the Rudné locality are in Table 4 and Table 5.

On the Belianske Lúky fen, there is the largest and most numerous locality of *Pedicularis sceptrum-carolinum*, the occurrence of which in Slovakia is limited to about 10 localities, and only in the foothills of the Tatras. There is also a vital and large population of relict *Carex limosa* and *Carex dioica* [23,54]. Based on a semi-quantitative analysis, 22 plant species were identified on the fen. Dominant species *Sphagnum palustre*, *Menyanthes trifoliata*, *Phragmites australis*, and *Filipendula ulmaria* were also found there. This type of peatland is one of the most abundant in the Central Europe. Species richness, as well as good nutrient tolerance, are characteristic of vascular plants [53,55]. At the edge of the fen, 14 species were identified with the dominant species *Galium verum* and *Cruciata laevipes*. On the pasture in the middle of agricultural land near the fen, we found 26 species with the dominant species *Trifolium pratense* and *Leucanthemum vulgare* (Table 4).

Based on the results of the Shannon index, we can state that the species diversity on the Belianske Lúky fen was low (2.4), whereas on the edge of the fen and on the permanent grassland near the fen, the value was low to medium (2.4 to 2.8). These findings are consistent with Davis and Anderson [45] who report low to high diversity on fens. The most balanced in terms of species were: the edge of the fen (0.86) → the neighboring plot (0.83) → the fen (0.79).

The Rudné Nature Reservation represents a unique bog of the highland type, and there are characteristic species of bryophytes and higher plants. *Ledum palustre*, *Andromeda polifolia*, *Drosera rotundifolia*, *Oxycoccus palustris*, *Vaccinium uliginosum*, *M. trifoliata*, and *Eriophorum* sp. occur in several bogs of the Upper Orava. *Empetrum nigrum*, *Rhynchospora alba*, and *Calla palustris* belong to rare species in Slovakia.

Five plant species have been identified on the Rudné bog. The dominant species are: *V. uliginosum* and *L. palustre*. At the edge of the bog, we identified six species, with the dominant species being *C. vulgaris*, *O. palustris*, and *C. limosa*. Ten species have been identified on a neighboring plot used as pasture, where the dominant species were *Rhinanthus serotinus*, *Trifolium hybridum*, *Taraxacum officinale*, *T. pratense*, *Dactylis glomerata*, and *Potentilla reptans* (Table 5). We recorded three legally protected and critically endangered species on the Rudné bog. The occurrence of *V. uliginosum* and critically endangered species *O. palustris*, *C. limosa*, and *L. palustre* from the Orava Basin is also reported by Bernátová and Mirga [56]. Similarly, Raučina and Janota [57] and Krištín [58] report the occurrence of *L. palustre*, *Calluna vulgaris*, and *V. uliginosum*. As shown in Table 5, we also confirmed the occurrence of the critically endangered species *C. limosa* outside the bog. Based on the results of the Shannon index, we can state that the species diversity on the Rudné bog was low (1.3) at the edge, and in the neighboring plot, the value was at the level of 1.2 to 1.7 i.e., low. The equitability values were in line with the diversity index. The most balanced in terms of species were: the bog (0.78) → the neighboring plot (0.72) → the edge of the bog (0.67).

The different structures of species composition and the occurrence of rare species are affected by several factors of these specific habitats [59,60]. The coverage of different plants in the studied areas can be explained by the different contents of N, P, and pH. Hájková and Hájek [61], Gagnon et al. [62] and Purre et al. [63] claim that pH is the main factor that creates the conditions for the development of various plants communities. A lower pH leads to the dominance of bog vegetation, while a higher pH is an assumption for vegetation similar to fens.

### 3.4. Hierarchical Cluster Analysis (HCA)

The dendrograms of hierarchical cluster analysis of the environmental parameters on the peatlands are presented in Figure 8. HCA identified three groups of associations in accordance with the monitored environmental parameters: Group I—BL f (Belianske Lúky fen); Group II—BL transition and BL soil; Group III—R b (Rudné bog), R transition, and R soil. The results of the HCA analysis in accordance with the grouping of monitored environmental parameters confirmed that there are significant differences between the studied peatlands. The fen of Belianske Lúky in comparison with the bog of Rudné has a very low OC content as well as the thickness of the peat horizon. Content of the Nt is higher in comparison with the bog of Rudné, which was reflected in a narrower C/N ratio, but at the same time, extremely low content of accessible P and an above-limit content of As in surface horizons were found. Significant differences were also found from a phytocoenological point of view. A total of 22 plant species was identified on BL f but only five on R b, which also affected the lower diversity (H’) and equitability (e) (Table 4 and Table 5).

### 3.5. Non-Metric Multidimensional Scaling (NMDS) Ordination Based on Bray–Curtis Dissimilarity of the Six Plots in the Peatland Complex

NMDS analysis of 14 environmental variables including OC, Nt, C/N, P, K, pH, As, Cd, Co, Cu, Cr, Ni, pH, and Zn, characterizing the six environments, which are the combination of two localities: BL f, BL transition, BL soil, R b, R transition, and R soil. The environments sharing the same locality were grouped with the same colors. Figure 9 shows that on the Belianske Lúky fen and outlying ecosystems, pH is the most important environmental variable that affects plant diversity, but on the Rudné bog, it is C/N, and the content of accessible P. Vector bearing is oriented in the direction of increasing values and the length is proportional to the strength of fit to the ordination. Chen et al. [64] confirmed in their study that the depth to the water table, redox potential, conductivity, and calcium were significant environmental factors associated with genus- and species-level composition. This study also suggested that a taxonomic analysis at the genus level may be a useful tool for assessing environmental changes in peatlands.

## 4. Conclusions

Peatlands are the highest reservoirs of terrestrial organic carbon and fulfil many ecological functions. This study presents the results of pedological and phytocoenological research focused on the detailed research of chemical parameters (pH, organic carbon, and nutrients), risk elements (As, Cd, Co, Cr, Cu, Ni, Pb, and Zn), and species composition of vegetation of two different peatlands on the territory of Slovakia: Belianske Lúky (a fen) and Rudné (a bog). The study revealed that the thickness of the peat horizon of the fen in comparison with the highland bog is very low. In terms of the quality of organic matter, the monitored peatlands are dominated by fresh plant residues, such as cellulose and lignin, and their degree of humic substances is relatively low. Differences between individual types of peatlands were also found in the other chemical parameters monitored, especially in the soil reaction and the supply of organic matter of peat with nitrogen. The values of the soil exchange reaction were neutral, or slightly alkaline on the fen, but extremely low on the bog. A significantly higher nitrogen supply was found in the organic matter of the fen, in contrast to the bog, the organic matter of which was poor in nitrogen. At the same time, extremely low content of accessible P and an above-limit content of As in the surface horizons were found on the fen. From the phytocoenological point of view, 22 plant species were identified on the fen, while only five species were identified on the bog, which also affected the higher diversity (*H’*) and equitability. The results of statistical testing (HCA and NMDS) confirmed the diversity of the studied peatlands and the different impact of environmental variables on plant diversity. Variables like pH, C/N, and P were important environmental factors associated with composition at the species level. As peat ecosystems are extremely sensitive to environmental changes (changes in groundwater levels, climate change), which can accelerate the aerobic mineralization of peats, it is necessary to carry out detailed monitoring at regular intervals. Based on changes in the concentration of organic carbon in peat deposits of the monitored localities, it will be possible to estimate the sequestration or elimination of organic carbon in peat ecosystems.

## Figures and Tables

**Figure 1 plants-10-01290-f001:**
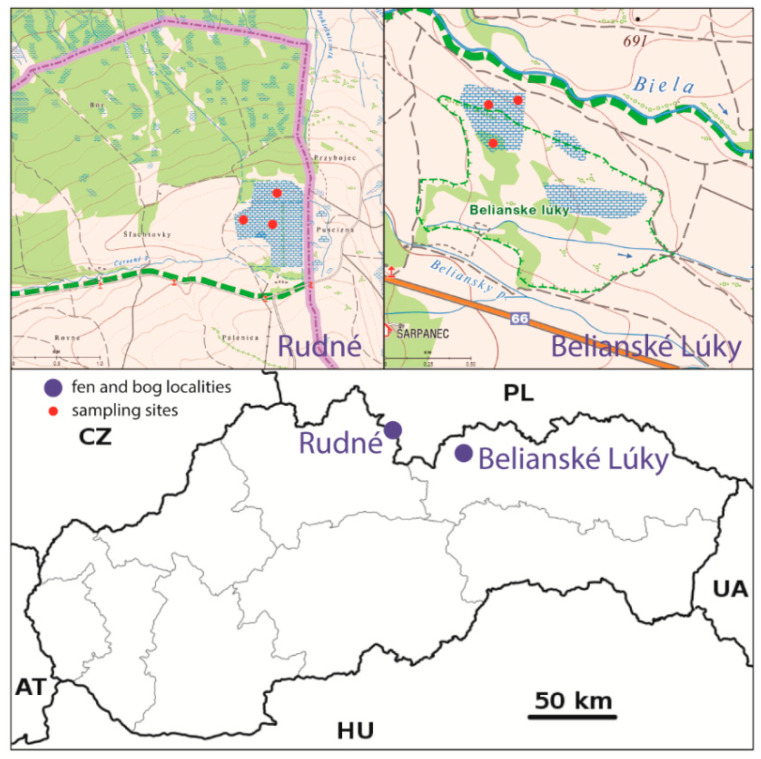
Location of monitored peatlands and sampling points.

**Figure 2 plants-10-01290-f002:**
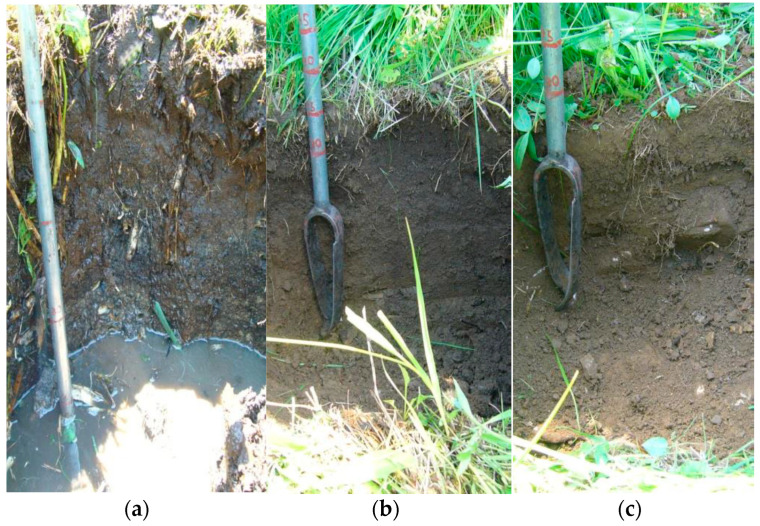
Profiles of sampling points at the Belianske Lúky locality. (**a**) Upper part of the peat profile, filled with water below 0.3 m; (**b**) Soil profile at the edge of the fen Rendzina organogenic; (**c**) Soil profile on the agricultural land Rendzina modal.

**Figure 3 plants-10-01290-f003:**
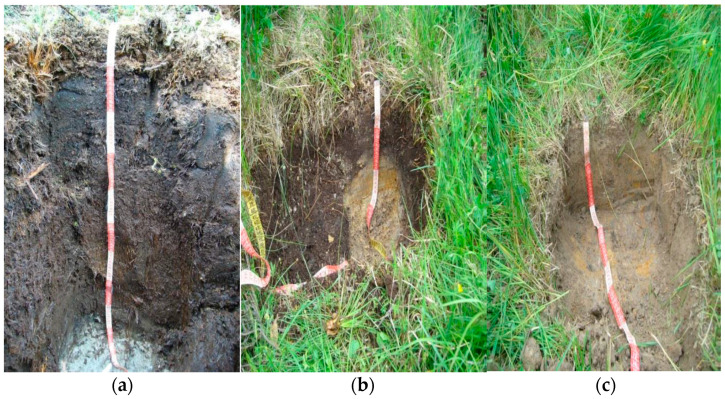
Profiles of sampling points at the Rudné locality. (**a**) Profile of the bog Organosol gleyic; (**b**) Soil profile at the edge of the bog—Cambisol pseudogleyic; (**c**) Soil profile on agricultural land Pseudogley modal.

**Figure 4 plants-10-01290-f004:**
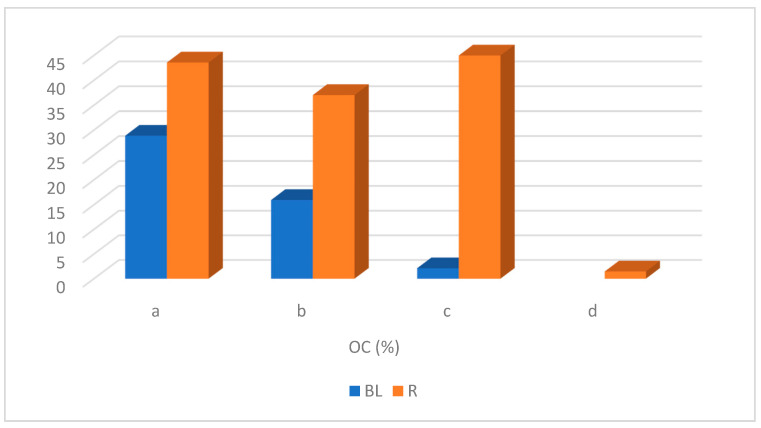
Organic carbon content (% OC) on peatlands Belianske Lúky and Rudné (BL—Belianske Lúky, a—5–15 cm, b—20–30 cm, c—110–120 cm; R—Rudné, a—0–10 cm, b—15–25 cm, c—35–45 cm, d—95–103 cm).

**Figure 5 plants-10-01290-f005:**
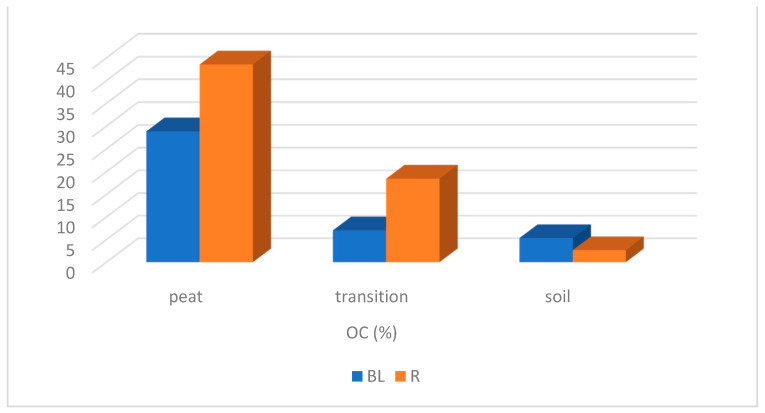
Organic carbon OC (%) values in the first depth of peatlands, transitions and soils (BL—Belianske Lúky, R—Rudné).

**Figure 6 plants-10-01290-f006:**
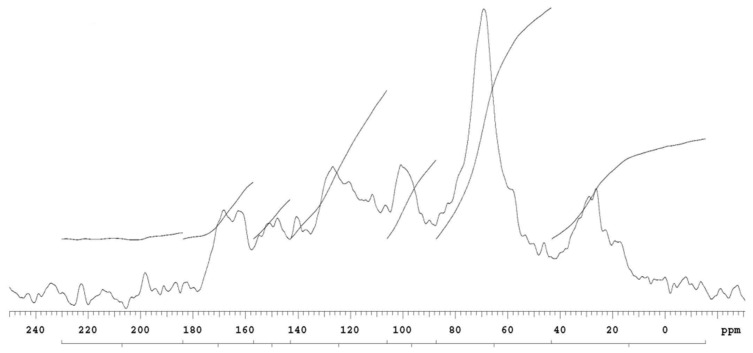
^13^C NMR the upper horizon spectra of the Rudné bog.

**Figure 7 plants-10-01290-f007:**
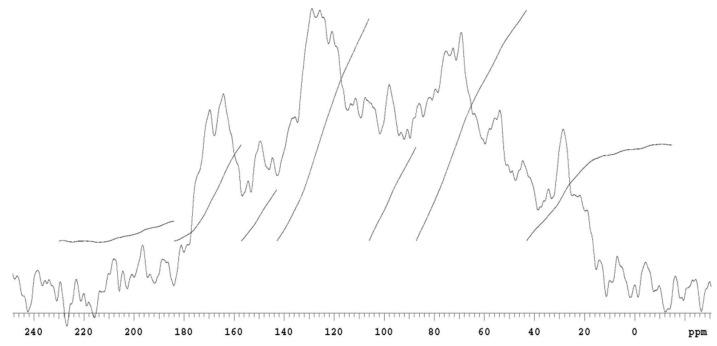
^13^C NMR the upper horizon spectra of the Belianske Lúky fen.

**Figure 8 plants-10-01290-f008:**
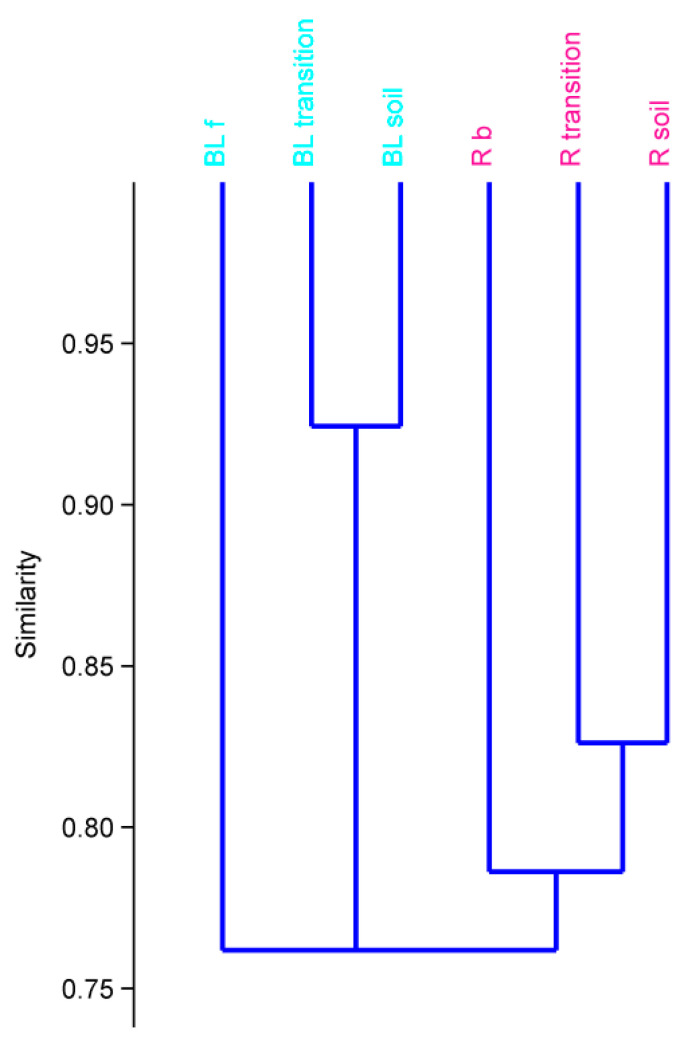
Dendrogram derived from hierarchical cluster analysis (HCA) of the environmental parameters on the peatlands (BL f—Belianske Lúky fen, R b—Rudné bog).

**Figure 9 plants-10-01290-f009:**
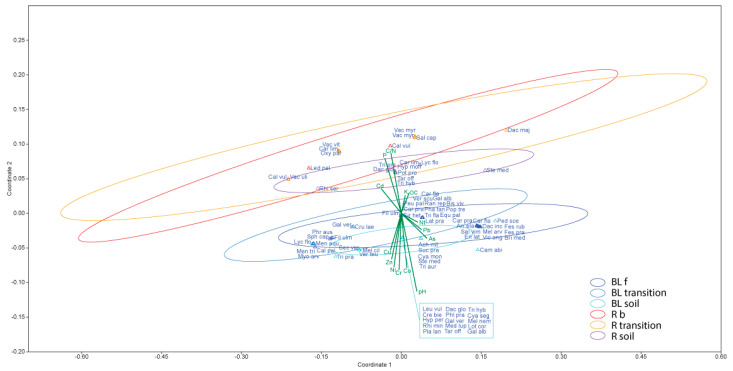
Graphic output of the NMDS ordination displaying localities and significant environmental vectors (BL f—Belianske Lúky fen, R b—Rudné bog).

**Table 1 plants-10-01290-t001:** Relative integrated intensities (% of total area) solid state ^13^C NMR of peat.

Peat	230–184	184–157	157–143	143–106	106–87	87–43	43–15	C_aliph_	C_ar_	α
	ppm							%		
BL f	2.44	11.86	6.31	27.45	11.62	28.44	11.88	51.94	33.76	39
R b	0.94	8.58	5.99	22.47	11.94	34.94	15.14	62.02	28.46	31

Notes: BL f—Belianske Lúky fen, R b—Rudné bog, C_alif_—15–106 ppm, C_ar_—106–157 ppm, α = C_ar_/(C_ar_ + C_alif_) × 100 (%).

**Table 2 plants-10-01290-t002:** Values of nutrients and OC (%), pH on peat (P), transition and soil of Belianske Lúky and Rudné.

Localities	Depth (cm)	OC (%)	Nt(mg/kg)	C/N	P (mg/kg)	K (mg/kg)	pH (KCl)
**BL f**	5–15	28.87	20,200	14.3	<0.4	377	7.3
**BL f**	20–30	15.96	9200	17.4	<0.4	132	7.6
**BL f**	110–120	2.12	1970	10.8	<0.4	84.9	7.4
**BL transition**	10–20	7.06	7300	9.7	<0.4	103	7.2
**BL soil**	10–20	5.36	5300	10.1	1.8	104	6.6
**R b**	0–10	43.57	14,300	30.5	94.5	636	3.0
**R b**	15–25	37.03	14,500	25.6	59.3	291	2.9
**R b**	35–45	44.98	12,000	37.0	17.9	61.9	2.8
**R b**	95–103	1.47	900	16.3	14.8	43.7	3.5
**R transition**	5–15	18.48	8400	22.0	3.5	180	3.4
**R soil**	3–13	2.66	2300	11.6	30.5	61.7	4.0

Notes: BL f—Belianske Lúky fen, R b—Rudné bog.

**Table 3 plants-10-01290-t003:** Values of risk elements (mg/kg) on fen and bog, transitions and soils of Belianske Lúky and Rudné.

Localities	Depth (cm)	As	Cd	Co	Cu	Cr	Ni	Pb	Zn
**BL f**	5–15	113.0	<0.2	2.3	4.4	9.5	0.0	58.6	71.0
**BL f**	20–30	64.9	0.4	3.5	2.3	4.1	0.1	25.1	46.6
**BL f**	110–120	7.3	<0.2	12.2	53.3	26.9	36.4	19.0	54.3
**BL transition**	10–20	8.4	0.3	<1.2	47.7	10.9	22.8	30.8	84.7
**BL soil**	10–20	13.3	0.4	6.3	54.0	17.8	35.1	29.6	94.6
**R b**	0–10	2.6	0.5	<1.2	2.2	9.8	1.9	28.6	110.1
**R b**	15–25	8.0	2.4	<1.2	6.7	9.4	3.1	88.6	61.5
**R b**	35–45	3.2	0.8	<1.2	2.5	2.8	0.7	52.3	37.5
**R b**	95–103	3.8	<0.2	10.9	27.3	4.8	6.3	13.4	14.7
**R transition**	5–15	5.5	0.6	<1.2	13.4	8.9	5.6	44.6	28.4
**R soil**	3–13	5.5	<0.2	<1.2	34.1	6.6	8.6	18.2	27.1
**Limit value ***		25	0.7	15	60	150	50	70	150

Notes: BL f—Belianske Lúky fen, R b—Rudné bog, * Act No. 220/2004 Coll. of Laws.

**Table 4 plants-10-01290-t004:** Phytocoenological data of peatlands of plant communities in the fen Belianske Lúky.

Localities	Covering (%)	Number of Species	Species	Code	*H*’	*e*
**BL f**	**100%**	**22**	**E_0_**: *Sphagnum palustre***4** **E_1_**: *Menyanthes trifoliata***5**, *Phragmites australis***4**, *Filipendula ulmaria***4**, *Dactylorhiza incarnata***2**, *Eriophorum latifolium***2**, *Salix viminalis***2**, *Carex flava***2**, *Carex x prahliana***2**, *Briza media***2**, *Arrhenatherum elatius***2**, *Festuca pratensis***2**, *Festuca rubra***2**, *Vicia angustifolia***2**, *Melampyrum arvense***2**, *Equisetum palustre***1**, *Populus tremula***1**, *Ranunculus reptans***1**, *Trisetum flavescens***1**, *Cirsium heterophyllum***1**, *Phacelia tanacetifolia***1**, *Lathyrus pratensis***1** **E_2_**, **E_3_**: no species were recorded	Sph pal Men tri Phr aus Fil ulm Dac inc Eri lat Sal vim Car fla Car pra Bri med Arr ela Fes pra Fes rub Vic ang Mel arv Equ pal Pop tre Ran rep Tri fla Cir het Pha tan Lat pra	**2.4**	**0.79**
**BL transition**	**100%**	**14**	****E_0_****: no species were recorded **E_1_**:*Galium verum***3**, *Cruciata laevipes***3**, *Veronica scutellaria***2**, *Bistorta vivipara***2**, *Galium album***2**, *Carex flacca***2**, *Carex x prahliana***2**, *Peucedanum palustre***2**, *Filipendula ulmaria***2**, *Mentha aquatica***1**, *Myosotis arvensis***1**, *Lychnis flos-cuculi***1**, *Caltha palustris***1**, *Pedicularis sceptrum- carolinum***r** ****E_2_****, **E_3_**: no species were recorded	Gal ver Cru lae Ver scu Bis viv Gal alb Car flc Car pra Peu pal Fil ulm Men aqu Myo arv Lyc flo Cal pal Ped sce	**2.3**	**0.86**
**BL soil**	**100%**	**26**	**E_0_**: no species were recorded **E_1_**: *Trifolium pratense***4**, *Veronica teucrium***3**, *Securigera varia***3**, *Melica ciliata***3**, *Leucanthemum vulgare***2**, *Dactylis glomerata***2**, *Galium verum***2**, *Hypericum perforatum***2**, *Medicago lupulina***2**, *Phleum pratense***2**, *Plantago lanceolata***2**, *Rhinanthus minor***2**, *Taraxacum officinale***2**, *Trifolium hybridum***2**, *Achillea millefolium***1**, *Cyanus montanus***1**, *Trifolium aureum***1**, *Stellaria media***1**, *Cyanus segetum***2**, *Succisa pratensis***1**, *Melampyrum nemorosum***2**, *Lotus corniculatus***2**, *Galium album***2**, *Crepis biennis***2**, *Campanula abietina+* ****E_2_****, **E_3_**: no species were recorded	Tri pra Ver teu Sec var Mel cil Leu vul Dac glo Gal ver Hyp per Med lup Phl pra Pla lan Rhi min Tar off Tri hyb Ach mil Cya mon Tri aur Ste med Cya seg Suc pra Mel nem Lot cor Gal alb Cre bie Cam abi	**2.8**	**0.83**

Notes: 5—cover of 75 to 100%, 4—cover of 50 to 75%, 3—cover of 25 to 50%, 2—cover of 5 to 25%, 1—under cover of 5%, +—cover of negligible, r—occasionally, BL f—Belianske Lúky fen.

**Table 5 plants-10-01290-t005:** Phytocoenological data of peatlands of plant communities in the bog Rudné.

Localities	Covering (%)	Number of species	Species	Code	*H*’	*e*
**R b**	**90%**	**5**	**E_0_**: no species were recorded **E_1_**: *Vaccinium uliginosum***5**, *Ledum palustre***4**, *Vaccinium vitis-idaea***3**, *Calluna vulgaris***2**, *Vaccinium myrtillus***1** ****E_2_****, **E_3_**: no species were recorded	Vac uli Led pal Vac vit Cal vul Vac myr	**1.3**	**0.78**
**R transition**	**80%**	**6**	**E_0_**: no species were recorded **E_1_**: *Calluna vulgaris***5**, *Oxycoccus palustris***3**, *Carex limosa***3**, *Vaccinium myrtillus***1**, *Salix caprea***1**, *Dactylorhiza majalis***r** **E_3_**: no species were recorded	Cal vul Oxy pal Car lim Vac myr Sal cap Dac maj	**1.2**	**0.67**
**R soil**	**100%**	**10**	**E_0_**: no species were recorded **E_1_**: *Rhinanthus serotinus***4**, *Trifolium hybridum***2**, *Taraxacum officinale***2**, *Trifolium pratense***2**, *Dactylis glomerata***2**, *Potentilla x procumbentireptans***2**, *Lychnis flos-cuculi***1**, *Carex limosa***1**, *Hypericum montanum***1**, *Stellaria media***+** ****E_2_****, **E_3_**: no species were recorded	Rhi ser Tri hyb Tar off Tri pra Dac glo Pot pro Lyc flo Car lim Hyp mon Ste med	**1.7**	**0.72**

Notes: 5—cover of 75 to 100%, 4—cover of 50 to 75%, 3—cover of 25 to 50%, 2—cover of 5 to 25%, 1—under cover of 5%, +—cover of negligible, r—occasionally, R b—Rudné bog.

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
