# Peer review of "Chemical and Phytocoenological Characteristics of Two Different Slovak Peatlands"

_plants, 2021, doi:10.3390/plants10071290_

Round 1

Reviewer 1 Report

The manuscript is very well written. The large number of authors is appropriate given the diverse talent that was required for soil morphology, lab analysis, statistical evaluation, plant taxonomy etc. Photos are a plus, figures / tables well done

Author Response

We would like to thank the opponent very much for the positive evaluation and the recommendation to publish the article. 

Reviewer 2 Report

The manuscript presents a detailed study of a fen and a bog in Slovakia, where measured parameters are pH, organic carbon including 13C-NMR spectroscopy, nutrients, trace elements (As, Cd, Co, Cr, Cu, Ni, Pb, and Zn), and plant species composition at three soil profiles at each location. The measurements seem to have been correctly performed and 13C-NMR is an advanced measurement technique. According to the authors, the aim is to identify indicator plant species of the two different peatlands and to examine the effects of the measured environmental variables for vegetation composition, diversity, and equitability. With only one bog and one fen to compare these conclusions cannot be drawn, several different bogs and fens has to be analyzed for this to get statistics. In my opinion a more proper and advanced scientific question would be preferable, the data is currently more of environmental monitoring than belonging in a scientific journal. The data is presented correctly most of the time, if not commented below. However, due to the lack of an advanced research question I recommend against publication at this stage. If it should be accepted, it also needs to be shortened; it is much text in Introduction but also in the Result section that is unnecessary and of very elemental knowledge, often also repeated. The text would be more suited for a textbook/report, than a scientific journal. Below are detailed comments that need to be addressed before publication.

Detailed comments:

“Heavy metals” is not considered a scientific term, e.g. As is included (a metalloid) and “heavy” is a bit arbitrary. Change to “risk elements” as is written at other places in the ms.

Decide upon an order of which site is written first (the bog or the fen) and be consistent throughout the whole paper, otherwise it becomes confusing.

Where duplicate samples used in the chemical analyses?

L160. What is the concentration of KCl?

L 195: Repetition from Introduction, remove sentence “Peat is characterized by more than 50% of combustible organic matter.”

L207-210. The different soil types is not an explanation of the different OC content. The soil type definitions are based on among other parameters the OC content, so it is a circular reasoning. Please provide another explanation involving the processes/mechanisms involved instead.

L 211-214. Fig. 4 It is a bit confusing to compare the sites, since the soil depths are different in all horizons. It would be better to show the data as a depth diagram, with depth on y-axis and organic carbon content on x-axis.

L220-223. Move to Introduction section and add papers from other research groups.

L 229. The term “humic acids” is less used recently (see e.g. Lehmann and Kleber, 2008)

L 238-240. Please reformulate this sentence, I do not understand.

L278-279. Is it certain that the one high Cd value is “real” and not due to contamination? Were duplicate samples analyzed?

L290 “tends to strongly bind heavy metals”. Reformulate. First of all, As is an metalloid, and it forms anions that is not as strongly bound to peat as cations are, most “heavy metals” form cations. Since the reference refer to As, use As instead, but I would not say that it binds strongly. Arsenic can certainly bind to peat, but it binds more strongly to more positive surfaces e.g. iron and aluminum (hydr)oxides. The amount of amorphous  Al and Fe (hydr)oxides should ideally have been analyzed using the acid oxalate extraction (Reeuwijk, 1995).

L 331. Please write which the three endangered species are in the Rudné bog.

L349 Table 4. Please move the lines with the summary to the top of each locality, now it is difficult to read. E.g. Move “BL f 100% 22 2.4 0.79 to the top on a separate line” and use bold font.

L399 humification. Please see comment above on L 229.

References

see Lehmann, J., Kleber, M., 2015. The contentious nature of soil organic matter. Nature 528, 60–68. https://doi.org/10.1038/nature16069

Reeuwijk, V. 1995. Procedures for Soil Analyses, sixth ed. (Wageningen, the Netherlands)

Author Response

Cover letter

We would like to thank our opponents very much for the valuable advice, comments, and recommendations that we tried to accept in full. We present our attitude to the justified comments in the chronological order.

Reviewer 2

The manuscript presents a detailed study of a fen and a bog in Slovakia, where measured parameters are pH, organic carbon including 13C-NMR spectroscopy, nutrients, trace elements (As, Cd, Co, Cr, Cu, Ni, Pb, and Zn), and plant species composition at three soil profiles at each location. The measurements seem to have been correctly performed and 13C-NMR is an advanced measurement technique. According to the authors, the aim is to identify indicator plant species of the two different peatlands and to examine the effects of the measured environmental variables for vegetation composition, diversity, and equitability. With only one bog and one fen to compare these conclusions cannot be drawn, several different bogs and fens has to be analyzed for this to get statistics. In my opinion a more proper and advanced scientific question would be preferable, the data is currently more of environmental monitoring than belonging in a scientific journal. The data is presented correctly most of the time, if not commented below. However, due to the lack of an advanced research question I recommend against publication at this stage. If it should be accepted, it also needs to be shortened; it is much text in Introduction but also in the Result section that is unnecessary and of very elemental knowledge, often also repeated. The text would be more suited for a textbook/report, than a scientific journal. Below are detailed comments that need to be addressed before publication.

Comments:

“Heavy metals” is not considered a scientific term, e.g. As is included (a metalloid) and “heavy” is a bit arbitrary. Change to “risk elements” as is written at other places in the ms.

- accepted

Decide upon an order of which site is written first (the bog or the fen) and be consistent throughout the whole paper, otherwise it becomes confusing.

- accepted

L160. What is the concentration of KCl?

- accepted, we added the concentration of KCl (L161-162).

L 195: Repetition from Introduction, remove sentence “Peat is characterized by more than 50% of combustible organic matter.”

- accepted

L207-210. The different soil types is not an explanation of the different OC content. The soil type definitions are based on among other parameters the OC content, so it is a circular reasoning. Please provide another explanation involving the processes/mechanisms involved instead.

- We disscused this issue with soil experts and renowned pedologists and our statement was confirmed by them.

L 211-214. Fig. 4 It is a bit confusing to compare the sites, since the soil depths are different in all horizons. It would be better to show the data as a depth diagram, with depth on y-axis and organic carbon content on x-axis.

- We mentioned reason of different sampling depth in Materials and Methods section. It is based on visual (morphological) changes in the soil profile.

L220-223. Move to Introduction section and add papers from other research groups.

- We don´t mention information about methods in the introduction, so we excluded this infromation.

L 229. The term “humic acids” is less used recently (see e.g. Lehmann and Kleber, 2008)

- accepted, it was replaced by „humic substances“

L 238-240. Please reformulate this sentence, I do not understand.

- accepted

L278-279. Is it certain that the one high Cd value is “real” and not due to contamination? Were duplicate samples analyzed?

- Samples were analyzed in triplicate. This is the average sample.

L290 “tends to strongly bind heavy metals”. Reformulate. First of all, As is an metalloid, and it forms anions that is not as strongly bound to peat as cations are, most “heavy metals” form cations. Since the reference refer to As, use As instead, but I would not say that it binds strongly. Arsenic can certainly bind to peat, but it binds more strongly to more positive surfaces e.g. iron and aluminum (hydr)oxides. The amount of amorphous  Al and Fe (hydr)oxides should ideally have been analyzed using the acid oxalate extraction (Reeuwijk, 1995).

- accepted, corrected

L 331. Please write which the three endangered species are in the Rudné bog.

- three endangered species are mentioned below in the text: Oxycoccus palustris, Carex limosa, and Ledum palustre

L349 Table 4. Please move the lines with the summary to the top of each locality, now it is difficult to read. E.g. Move “BL f 100% 22 2.4 0.79 to the top on a separate line” and use bold font.

- accepted

L399 humification. Please see comment above on L 229.

- accepted

Reviewer 3 Report

The manuscript entitled “Chemical and Phytocoenological Characteristics of two Different Slovak Peatlands” has been well structured and has merit in the presentation of data. All the results were presented meticulously and defended by relevant data and figures. It would appeal to the audience of the journal.

The authors have employed all standard tools and methods. However, the following queries should be incorporated before further consideration. Comments are as follows:

  • Authors should avoid to use terms like “Our study line 21. Correct throughout the manuscript.
  • Why did the authors not add a section on statistical analysis? “A brief section on statistical analysis should be included. 
  • Figure 4 & 5. Units should be mentioned on the y-axis, and standard deviation/error or a standard error has not been included in the statistical analysis?  
  • Standard deviation/error should also add in tables.
  • In discussion: authors should add why only the selected heavy metals were chosen in this study; justification should be included in the discussion. Also, discuss considering recent references justifying the presence of these heavy metals in the soil.
  • Section: 3.3. Phytocoenological Characteristics; Authors should properly follow the rules of abbreviation use of botanical names? For the first time- Ledum palustre then L. palustre 
  • Section: 3.4. Hierarchical Cluster Analysis (HCA): Please mention the Euclidean distance in HCA. 
  • Line 410: pH, C/N, and P were important environmental factors associated with composition at the species level. Authors should avoid the begin sentence by symbols/abbreviations. Pls, check throughout the manuscript. 
  • The conclusion is not well presented. It included the introductory part, experimental details but main findings and recommendations of a study that should be added in conclusion. Rewrite the conclusion considering significant results. 
  • There are several unclear sentences. Language should be checked carefully.

Author Response

Cover letter

We would like to thank our opponents very much for the valuable advice, comments, and recommendations that we tried to accept in full. We present our attitude to the justified comments in the chronological order.

Reviewer 3

The manuscript entitled “Chemical and Phytocoenological Characteristics of two Different Slovak Peatlands” has been well structured and has merit in the presentation of data. All the results were presented meticulously and defended by relevant data and figures. It would appeal to the audience of the journal.

The authors have employed all standard tools and methods. However, the following queries should be incorporated before further consideration. Comments are as follows:

Comments:

Authors should avoid to use terms like “Our study line 21. Correct throughout the manuscript.

- accepted, we corrected it throughout the manuscript

Why did the authors not add a section on statistical analysis? “A brief section on statistical analysis should be included.

-Statistical analysis is included in sections 3.4 and 3.5.

Figure 4 & 5. Units should be mentioned on the y-axis, and standard deviation/error or a standard error has not been included in the statistical analysis? Standard deviation/error should also add in tables.

- Figure 4 & 5 and tables are not the results of statistical analysis.

In discussion: authors should add why only the selected heavy metals were chosen in this study; justification should be included in the discussion. Also, discuss considering recent references justifying the presence of these heavy metals in the soil.

- We selected risk elements based on Act No. 220/2004 Coll. of Laws.

- We stated brief disscusion about risk elements throughout the manuscript in section Results.

Section: 3.3. Phytocoenological Characteristics; Authors should properly follow the rules of abbreviation use of botanical names? For the first time- Ledum palustre then L. palustre

- accepted

Section: 3.4. Hierarchical Cluster Analysis (HCA): Please mention the Euclidean distance in HCA.

  • We used The Bray Curtis dissimilarity to quantify the differences between two different sites. It’s used primarily in ecology and biology. Euclidean distance, although not suitable for our ecological data. When comparing two samples, Euclidean distance puts more weight on differences in species abundances than on difference in species presences. As a result, two samples not sharing any species could appear more similar (with lower Euclidean distance) than two samples which share species but the species largely differ in their abundances.  

Line 410: pH, C/N, and P were important environmental factors associated with composition at the species level. Authors should avoid the begin sentence by symbols/abbreviations. Pls, check throughout the manuscript.

- accepted, we checked it throughout the manuscript and fixed it

The conclusion is not well presented. It included the introductory part, experimental details but main findings and recommendations of a study that should be added in conclusion. Rewrite the conclusion considering significant results. There are several unclear sentences. Language should be checked carefully.

- In our opinion we stated the most important findings and recommendations in Conclusion section.

Round 2

Reviewer 2 Report

The authors have replied satisfactorily on all my comments, so this version can be accepted for publication.

Reviewer 3 Report

The authors responded to all my remarks and the manuscript can be accepted.